# Predictors of food variety and food consumption scores of adolescents living in a rural district in Ghana

**Michael Akenteng Wiafe**[1]*, **Jessica Ayensu**[2], **Georgina Benewaa Yeboah**[3]

**1** Department of Nutritional Sciences, University for Development Studies, Tamale, Ghana, **2** Department of Clinical Nutrition and Dietetics, University of Cape Coast, Cape Coast, Ghana, **3** School of Food and Biological Engineering, Jiangsu University, Zhenjiang, Jiangsu, 212013, China

* mawiafe5@gmail.com

## Abstract

### Introduction

There is a dearth of information about the food variety and consumption scores of adolescents in Ghana. This study assessed predictors of food variety and consumption scores of adolescents living in a rural district in Ghana.

### Method

In this cross-sectional study, a multi-stage sampling method was used to select one hundred and thirty-seven (137) adolescents from the Asante-Akim South Municipality of Ghana. A structured questionnaire was used to collect data on sociodemographic, food practices and dietary intakes of study participants. Descriptive, chi-square, T-test, partial correlation and binary logistic regression were used for the data analysis.

### Results

The mean food variety score was 25.8±6.4 (range 7–42) and food consumption score was 35±5.1 (range 18.6–49.9). Food variety score was significantly ($p<0.05$) associated with guardian income status. A significant and positive partial correlation existed between food variety score and calcium intake ($r = 0.236$, $p<0.05$). About 49% and 51% of adolescents were food insecure and food secure, respectively. Food consumption score had significant association ($p<0.05$) with gender ($X^2 = 6.1$), residence ($X^2 = 7.0$), frequency of meal ($X^2 = 6.8$) and food variety score ($X^2 = 5.4$). Adolescent male (AOR = 2.3, 95% CI (1.2–4.6), p = 0.017), peri-urban residency (AOR = 2.1, 95%CI (1.0–4.4), p = 0.036), having three or more meals per day (AOR = 4.2, 95% CI (1.3–13.6), p = 0.018), and a high food variety score (AOR = 2.1, 95%CI (1.0–4.2), p = 0.041) significantly predicted food consumption scores.

### Conclusion

Moderate income status was associated with food variety score in our study participants. Linear relationship existed between food variety and calcium intake. Adolescent males,

**Data Availability Statement:** The data used to support the findings of this study may be assessed by writing to the Chairman Committee on Human Research Publication Ethics, Room 8 Anatomy Block 3, School of Medical Sciences, Kwame

Nkrumah University of Science and Technology, Kumasi, Ghana, or chrpe.knust.kath@gmail.com.

**Funding:** The author(s) received no specific funding for this work.

**Competing interests:** The authors have declared that no competing interests exist.

peri-urban residency, frequency of meals and high food variety score were the predictors of food consumption score. Nutrition-specific and nutrition sensitive programmes aimed at promoting adolescent health should involve guardians.

## Introduction

Adolescence is the period in the stages of human development when the growth rate of the human body is very rapid and individuals acquire a substantial proportion of their adult weight and height [1]. The consumption of a poor quality diet among adolescents could consequently affect generations as adolescence is a sensitive period for the development of both productive and reproductive capacities [2,3]. Suboptimal nutrition may predispose adolescents to the consequences of poor diet in adulthood. As no single food group contains all the nutrients required for adolescents' growth and development, this necessitates the need for adolescents to eat from a variety of sources of food and have acceptable food security.

The food variety score measures the number of distinct foods eaten over a designated period [4]. This is used for the evaluation of the quality of dietary patterns [5]. Variety in essence, is choosing to eat a mixture of foods from different food groups and a mixture from within these food groups. The consumption of a varied diet improves nutritional adequacy, encourages biodiversity and sustainability and minimizes the adverse consequences of food on health, while improving the overall health of an individual [6]. The consumption of a varied diet in the appropriate quantity is important for optimal growth and development, as well as the total wellbeing through all the cycles of life, including adolescence [7]. Food variety was negatively associated with food insecurity, food neophobia, and sensory motive while semi-urban areas, urban areas, meal planning practices and high income had a positive association with food variety [7–9]. Increasing food variety contributed to improved diet quality among adolescents living with type 1 diabetes [10]. Thus, the consumption of a wide array of foods during adolescence cannot be overemphasized, as these individuals are especially prone to malnutrition originating from the increased nutritional needs for optimum growth and development.

A high food variety score has been linked to higher education, more frequent meals, urban residence, adequate micronutrients intake and healthy bone growth [11–13]. A positive and significant relationship has been found between high food variety score and serum ferritin, serum magnesium and vitamin C intakes whiles a negative association was reported to exist between a high food variety score and low intake of sugar, salt and saturated fat [5,14]. However, a low food variety score was observed among young people and has been associated with physical frailty in adulthood [14,15]. Therefore, if adolescents would eat from a wide variety of food groups, this could serve as a cost-effective approach with the potential to promote health, and minimize the occurrence of non-communicable diseases as they age or grow, since this period sets the precedent for health in adulthood. Food variety score had a positive association with dietary micronutrients and an inverse relationship with inadequate micronutrients intakes [16]. These micronutrients are needed to support the physiological and physical developmental processes in adolescents.

The food consumption score is an indicator of how a household or individual is food secured or not. It is based on dietary diversity, food frequency, and the relative nutritional relevance of the distinct food groups [17]. Residency, high-income level, income diversification, high education level, animal source of foods, good dietary practices and religion positively influenced food consumption scores [18–26].

Adolescents in food secure households are less likely to suffer nutritional deficiencies, the detrimental effect of undernutrition on cognitive development, and mental health disorders such as poor mood, anxiety, depression, and behavioural issues [27,28]. Food insecurity has also been documented to pose health challenges such as poor dietary pattern, obesity, asthma and increased the health care cost for adolescents [29–31]. This health issues persist into adulthood [32].

In Ghana, 'adolescent nutrition' receives very little attention, even though this stage of life is critical for development and health. The food variety and the food consumption score, are important indicators of nutritional adequacy and health status. There is however, a dearth of research on the food variety and consumption scores of adolescents. This study will provide information about food variety scores, food consumption scores and its predictors among adolescents. Thus, the study sought to identify predictors of the food variety and the food consumption scores of adolescents living in a rural district in Ghana.

## Materials and methods

### Study design, area and participants

This cross-sectional study was conducted at the Asante-Akim South municipality in the Ashanti Region of Ghana. It is one of Ghana's rural districts where farming is the main form of employment. The district has land size of 0.5%, out of the total land area of Ghana. The food crops largely cultivated are cassava, plantain, cocoyam, yam, rice and maize. About 83% of the population lives in the rural areas of the district while rest reside within the peri-urban zone of the district [33,34].

A total of one hundred and thirty-seven (137) healthy adolescents aged 10–14 years were selected using a multi-stage sampling method from the municipality for this study. Adolescents who were sick, on a special diet, on medication or had a chronic condition were excluded from the study.

### Data collection

Sociodemographic, food practices, and dietary intake data were collected with the aid of a structured questionnaire.

### Food practices

Adolescents provided information about the frequency of meals they had per day and their meal skipping practices.

### Dietary intake

Dietary information for three days (one weekend and two working days) was gathered using a 24-hour recall. Participants were asked to tell us how many times (breakfast, mid-morning snack, lunch, mid-afternoon snack, dinner and bed-time snack) they ate a day and the detailed (composition) of the food consumed. Food handy measures were used to estimate how much of the food consumed at each meal. Each food was converted to grams and the nutrients were estimated using the Nutrient Analysis Template developed by the University of Ghana. The averages for the nutrients were recorded.

A validated food frequency questionnaire (FFQ) was used to assess the number of times participants consumed cereals, bread and other staples, rice and pasta dishes, starchy roots and plantain, pulses and lentils, nuts, meat, fish and poultry, dairy products fats and oils, soups and

stews sauces, vegetables (fresh, frozen and canned), fruits–fresh, alcoholic and non-alcoholic beverages, and sweets and snacks over a period of seven days.

### Food variety score

The food variety score was calculated from 63 selected food items from the FFQ. The list of foods from alcoholic and non-alcoholic beverages (such as 'pito', palm wine and coffee), sweets and snacks were excluded from the calculation of the food variety scores. The list of foods exempted were rarely or never consumed. Food item consumed at least once a week were scored '1' and '0' for those not consumed at all. The scores were summed to obtain the food variety score for each participant. The food variety score (FVS) was ranked 'high' ($\geq$26) or 'low' ($\leq$25) based on the mean food variety scores.

### Food consumption score

The foods on the FFQ list were further categorised into eight food groups: starchy staples; pulses; fruits; vegetables; meat, fish and poultry; dairy products; sugary products and edible oils. The frequency of the foods consumed in the same food group was summed. Each food group was multiplied by a weight factor: fruits and vegetables (1); starchy staples (2); pulses (3); meat, fish and poultry products (4); dairy products (4) and sugary and edible oils (0.5) [17]. The weight of the composite food groups was summed to obtain the food consumption score (FCS) of participants. The food consumption scores were classified into food insecure (food insecurity) (poor: 0–21; borderline: 21.5–35) and food secure (food security) (acceptable: >35) [17,21].

### Statistical analysis

A statistical package for social sciences (SPSS version 25) was used for the data analysis. Data were presented as means, standard deviation, frequencies and percentages. Descriptive statistics were used in the analysis of food consumption scores and food variety scores. Chi-square and T-test were used to test for the association between sociodemographic, food variety score and food consumption status. Binary logistics regression was used to predict factors related to food security. Partial correlation (r) was used to assess the relationship food variety score, food consumption score and micronutrients. All p-values were significant at p<0.05.

## Results

### Descriptive statistics

Out of the 137 adolescents who participated in the study, 51.1% were males and 48.9% were females. In terms of residency, 50.4% lived in peri-urban areas and 49.6% lived in rural communities. Majority of participants were in primary school (70.1%) and few (29.9%) at the Junior High School (JHS). Guardians with formal education were 78.1% and those with non-formal education constituted 21.9%. About 14% of adolescents took two meals per day, while 86% took three or more. The affirmative response on meal skipping was 55.5% and the negative was 44.5% (Table 1).

On the account of food variety score, adolescents with guardians in the moderate-income category (29.4±4.9) had higher mean food variety scores than those from the low-income category (24.5±6.3) (p<0.001). Females (26.9±6.3) had higher mean food variety scores compared to males (24.8±6.2) (p = 0.050). A higher mean food variety score was also recorded among peri-urban settlers (26.1±7.3), JHS students (26.9±4.5), guardians with formal education (26 ±6.1) and the consumption of three or more meals per day by study participants (25.9±6.4).

**Table 1. Sociodemographic, food practices, food variety score and food consumption score of adolescents.**

| Variable | N (%) | Food variety score Mean ± Sd p-value | Food consumption score Food insecure Food secure $X^2$ p-value |
|---|---|---|---|
| **Sex** | | | |
| Male | 70(51.1) | 24.8±6.2 0.050 | 27(40.3) 43(61.4) 6.1 0.017 |
| Female | 67(48.9) | 26.9±6.3 | 40(59.7) 27(38.6) |
| **Community** | 69(50.4) | 26.1±7.3 0.573 | 26(38.8) 43(61.4) 7.0 0.010 |
| Peri-urban | 68(49.6) | 25.5±5.2 | 41(61.2) 27(38.6) |
| Rural | 41(29.9) | 26.9±4.5 0.122 | 23(34.3) 18(25.7) 1.2 0.351 |
| **Educational status** | 96(70.1) | 25.3±7.0 | 44(65.7) 52(74.3) |
| **Participants** | 107(78.1) | 26.0±6.1 0.444 | 52(77.6) 55(78.6) 0.0 1.000 |
| JHS | 30(21.9) | 24.9±7.2 | 15(22.4) 15(21.4) |
| Primary | 100(73.0) | 24.5±6.3 <0.001 | 47(70.1) 53(75.7) 0.5 0.564 |
| **Guardian** | 37(27.0) | 29.4±4.9 | 20(29.9) 17(24.3) |
| Formal education | 19(13.9) | 25.3±6.4 0.692 | 4(6.0) 15(21.4) 6.8 0.012 |
| Non-formal education | 118(86.1) | 25.9±6.4 | 63(94.0) 55(78.6) |
| **Guardian income status** | 76(55.5) | 26.0±6.8 0.634 | 42(62.7) 34(48.6) 2.8 0.122 |
| Low (<Ghc490) | 61(44.5) | 25.5±5.9 | 25(37.3) 36(51.4) |
| Moderate (Ghc500-1000) | 65(47.4) | 20.9±4.7 <0.001 | 25(37.3) 40(57.1) 5.4 0.026 |
| **Frequency of meal per day** | 72(52.6) | 30.3±3.9 | 42(62.7) 30(42.9) |
| Two | | | |
| Three or more | | | |
| **Meal skipping** | | | |
| Yes | | | |
| No | | | |
| **Food variety status** | | | |
| Low | | | |
| High | | | |

Junior High School (JHS), Chi-square ($X^2$), Fisher's exact test, p<0.05, Frequency (N), percentage (%).

The highest food variety score was 52.6% and the lowest food variety score was 47.4% (Table 1).

A large proportion of participants in the food secure category had primary education (74.3%), guardian with formal education (78.6%), low-income status (75.7%) and did not skip meals (51.4%). Nonetheless, none of these factors had a significant association with food security (p>0.05). The study showed that more males (61.4%) relative to females (38.6%), were food secured ($X^2$ = 6.1, p = 0.017). The participants with peri-urban residential status (61.4%) were more likely to be food secure compared to those in the rural areas (38.6%) ($X^2$ = 7, p = 0.010). Food security was significantly associated with the frequency of meals per day ($X^2$ = 6.8, p = 0.012) (three or more -78.6%; two meals -21.4% (Table 1).

## Results

### Food variety score and food consumption score of participants

Table 2 shows the mean food variety score and food consumption score of participants. The results showed that the mean food variety score was 25.8±6.4, with a minimum and maximum score of 7 and 42, respectively. The mean food consumption score was 35±5.1, with a minimum and maximum score of 18.6 and 49.9, respectively. Adolescents with poor (0.7%) and borderline (48.2%) food insecurity totalled 48.2%. More than half (51.1%) of the participants had acceptable food security status (FCS >35).

### Predictors of food security

The binary logistics regression for predictors of food consumption status is presented in Table 3. After adjusting for age and/or sex, the outcome indicated that being an adolescent

**Table 2. Food variety score and food consumption score of participants.**

| Variables | Mean ± Sd min—max N (%) |
|---|---|
| Food variety score | 25.8±6.4 7.0–42.0 |
| Food consumption score | 35±5.1 18.6–49.9 |
| **Food consumption status** | 1(0.7) |
| **Food insecure** | 66(48.2) |
| Poor | 70(51.1) |
| Borderline | |
| **Food secure** | |
| Acceptable | |

Standard deviation = Sd, minimum (min), maximum (max), Frequency (N), Percentage (%).

male (AOR = 2.3, 95% CI (1.2–4.6), p = 0.017), living in a peri-urban community (AOR = 2.1, 95%CI (1.0–4.4), p = 0.036) taking three or more meals per day (AOR = 4.2, 95%CI (1.3–13.6), p = 0.036), and having a high food variety score (AOR = 2.1, 95%CI (1.0–4.2), p = 0.041) had higher odds of being food secured.

## Partial correlation of food variety and food consumption scores and dietary micronutrient intake

After adjusting for age and sex, a positive partial correlation was found between food variety score and dietary micronutrients intakes except for Vitamin A and Vitamin $B_{12}$. A significant relationship existed between food variety score and calcium (r = 0.236, p<0.05) (Table 4).

The partial correlation showed a weak positive relationship between food consumption score and most of the dietary vitamins and minerals. An inverse weak correlation was found between food consumption score and pyridoxine, vitamin C and sodium. None of the relationship was significant (p<0.05) (Table 4).

## Discussion

Food variety and food consumption scores of adolescents are critical to their health and well-being as it has a significant effect on adulthood and the unborn generation. Adolescents eating from a variety of food groups have the potential to increase nutrient adequacy. The study assessed the predictors of food variety and consumption scores of adolescents living in a rural

**Table 3. Binary logistics regression for predictors of food consumption status.**

| Predictors | Food Secure | |
|---|---|---|
| | Unadjusted β OR(95%CI) p-value | Adjusted β OR(95%CI) p-value |
| **Sex** | | |
| Male | 0.858 2.4(1.2–4.7) 0.014 | 0.836 2.3(1.2–4.6) 0.017 |
| Female | 1 | 1 |
| **Community** | 0.921 2.5(1.3–5.0) 0.009 | 0.760 2.1(1.0–4.4) 0.036 |
| Peri-urban | 1 | 1 |
| Rural | 1.458 4.3(1.3–13.7) 0.014 | 1.427 4.2(1.3–13.6) 0.018 |
| **Frequency of meals per day** | 1 | 1 |
| Thrice or more | 0.806 2.2(1.1–4.4) 0.021 | 0.731 2.1(1.0–4.2) 0.041 |
| Twice | 1 | 1 |
| **Food variety status** | | |
| High | | |
| Low | | |

β = Beta coefficient, Odds ratio (OR), p<0.05, After adjusting for age and/or sex.

**Table 4. Partial correlation of food variety and food consumption scores and dietary micronutrient intake.**

| Variable | Food variety score | Food consumption score |
|---|---|---|
| Food variety score | | 0.016 |
| Food consumption score | 0.016 | |
| Vitamin A | -0.009 | 0.109 |
| Thiamine | 0.114 | 0.063 |
| Riboflavin | 0.060 | 0.001 |
| Niacin | 0.144 | 0.103 |
| Pyridoxine | 0.081 | -0.021 |
| Vitamin C | 0.054 | -0.119 |
| Folate | 0.118 | 0.107 |
| Vitamin $B_{12}$ | 0.008 | 0.001 |
| Iron | 0.156 | 0.012 |
| Zinc | 0.151 | 0.058 |
| Calcium | 0.236* | 0.135 |
| Sodium | 0.158 | -0.043 |

Controlled variables: Age and sex, *Partial correlation is significant at the 0.05 level (2-tailed).

district in Ghana. The results showed that guardian income status was significantly associated with food variety scores. Food variety score was significantly related to dietary calcium intake. Sex, residence, frequency of meals per day and food variety score were significantly associated with food consumption scores. Food security was significantly related to adolescent males, peri-urban residency, three or more meals per day and high food variety score.

The current study showed a significant association between food variety scores and adolescents living with guardians of moderate-income status. The income level of a household influences the availability of food, the type of foods, nutritional quality of the food and dietary intake [35]. Adolescents from these households also had higher mean food variety scores. This implies that these participants had the wherewithal to purchase a wide range of foods and therefore ate from different sources of food. A study indicated that high income households had the opportunity to purchase and consume from wide sources of food [36]. Our finding is consistent with a study conducted among adolescents in Ethiopia [7]. The majority of adolescents in this study had a higher food variety score. This outcome is contrary to Drewnowski et al. [14] who worked with young people (20–30 years). The differences might be brought on by the age disparities between the research populations, the busy work schedules of young adults compared to adolescents, and the geographical locations. A significant and positive linear relationship existed between food variety score and dietary calcium intake. Dietary variety improved intake of micronutrients such as calcium, vitamin C and vitamin $B_2$ [37]. There was also a linear relationship between food variety score and dietary micronutrient such as thiamine, niacin, folate, iron, zinc and sodium. The outcome aligns with other studies that established that food variety scores had a positive relationship with micronutrient [5,16]. The ability to eat from a wide range of foods will increase the chances of consuming several micronutrient in the diet, as the various foods within the groups, might contain different micronutrient.

The study outcome on food consumption scores showed a significant association between sex, residency, frequency of meals per day and food variety score. The association between residency and the food consumption score is confirmed by a study among pregnant women in Ethiopia [21]. Results of a recent study in Burkina Faso also supports the findings that rural or urban settlement influences food consumption [20]. One's geographical location affects the

type of food items that can be grown, be available or be accessible, and these can influence the varieties of food that may be available to them.

Food consumption scores were used as indicators of food security, the higher the score, the more food secure the person is. The majority of the participants had acceptable food consumption scores (food security). This outcome is similar to that of Kong et al. [19]. Findings from this study showed that adolescent males, peri-urban settlement, consumption of three or more meals per day and high food variety scores had higher odds for food security. Most of the rural settlers send their produce to urban and peri-urban areas for sale on market days, and this could have contributed to the increased food security situation among participants living in the peri-urban areas. The higher number of participants being food secure may also be attributed to the high food variety scores and the educational status of guardians. A study in Zambia documented that, household heads with high educational status positively contributed to food security in the household [25]. Furthermore, a study in the Northern part of Ghana indicated that improved education (higher formal education) led to high food security status in the adult population [22].

## Conclusion

The study showed that majority of adolescents had acceptable food consumption scores and high food variety scores. The income status of guardian had significant impact on the food variety score of adolescents. Factors such as gender, residency, frequency of meals per day and food variety score were significantly associated with food consumption scores. The predictors of food security based on the study outcomes were adolescent males, peri-urban residency, three or more frequency of meals per day and high food variety score. Government and non-governmental organizations are encouraged to implement nutrition education programmes and interventions aimed at providing adolescents with adequate nutrition knowledge, and also empower guardians economically to increase their purchasing power as it will influence their wards to have access to variety of foods for consumption and possibly increase their food consumption scores. A well-nourished adolescent is a potential healthy adult. Nutrition-specific and nutrition sensitive programmes aimed at promoting adolescent health should involve guardians.

As a limitation to this study, the recall bias of the dietary intake methodology could affect the food consumption and variety scores. The subjective weight factor of the food consumption score may not be applicable for all food consumption patterns in all countries. The food consumption score is not able to show the seasonal changes in food and also measure the quantity of food gap.

## Ethics

The study protocol was approved by the Ethics Review Board of the Kwame Nkrumah University of Science and Technology, (reference number: CHRPE/AP/585/22), Ghana. Permission was granted by the Asante-Akim South Municipal Health Directorate. The purpose of the study was explained to the adolescents and guardians. Adolescents who gave their assent backed by their guardian's consent either by signing or thumb printing on the informed consent form qualified to participate in the study.

## Acknowledgments

Research assistants, adolescents and guardian are acknowledged.

## Author Contributions

**Conceptualization:** Michael Akenteng Wiafe, Jessica Ayensu.

**Data curation:** Michael Akenteng Wiafe, Georgina Benewaa Yeboah.

**Formal analysis:** Michael Akenteng Wiafe, Jessica Ayensu, Georgina Benewaa Yeboah.

**Investigation:** Michael Akenteng Wiafe, Jessica Ayensu.

**Methodology:** Jessica Ayensu, Georgina Benewaa Yeboah.

**Project administration:** Michael Akenteng Wiafe.

**Supervision:** Michael Akenteng Wiafe, Jessica Ayensu.

**Validation:** Michael Akenteng Wiafe, Georgina Benewaa Yeboah.

**Visualization:** Jessica Ayensu, Georgina Benewaa Yeboah.

**Writing – original draft:** Michael Akenteng Wiafe, Georgina Benewaa Yeboah.

**Writing – review & editing:** Michael Akenteng Wiafe, Jessica Ayensu, Georgina Benewaa Yeboah.

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
