## [Decision Letter · Decision Letter 0]

20 Dec 2022

PONE-D-22-28451PREDICTORS OF FOOD VARIETY AND FOOD CONSUMPTION SCORES OF ADOLESCENTS LIVING IN A RURAL DISTRICT IN GHANAPLOS ONE

Dear Dr. Wiafe,

Thank you for submitting your manuscript to PLOS ONE. After careful consideration, we feel that it has merit but does not fully meet PLOS ONE’s publication criteria as it currently stands. Therefore, we invite you to submit a revised version of the manuscript that addresses the points raised during the review process.

Please note that we have only been able to secure a single reviewer to assess your manuscript. We are issuing a decision on your manuscript at this point to prevent further delays in the evaluation of your manuscript. Please be aware that the editor who handles your revised manuscript might find it necessary to invite additional reviewers to assess this work once the revised manuscript is submitted. However, we will aim to proceed on the basis of this single review if possible.

We look forward to receiving your revised manuscript.

Kind regards,

Callam Davidson

Editorial Office

PLOS ONE

Journal Requirements:

4. Please ensure that the study is reported according to the STROBE guideline, and include the completed STROBE checklist as Supporting Information. Please add the following statement, or similar, to the Methods: "This study is reported as per the Strengthening the Reporting of Observational Studies in Epidemiology (STROBE) guideline (S1 Checklist)."

Did your study have a prospective protocol or analysis plan? Please state this (either way) early in the Methods section.

The terms gender and sex are not interchangeable (as discussed in https://www.who.int/health-topics/gender); please use the appropriate term.

Sample sizes must be large enough to produce robust results, where applicable. We are concerned that the manuscript does not include details of how the sample size was determined. It is therefore unclear whether the sample size employed is appropriate in light of the study design and manuscripts’ conclusions.

Reviewers' comments:

Reviewer's Responses to Questions

**Comments to the Author**

1. Is the manuscript technically sound, and do the data support the conclusions?

Reviewer #1: Yes

2. Has the statistical analysis been performed appropriately and rigorously? 

Reviewer #1: Yes

3. Have the authors made all data underlying the findings in their manuscript fully available?

Reviewer #1: Yes

4. Is the manuscript presented in an intelligible fashion and written in standard English?

Reviewer #1: Yes

5. Review Comments to the Author

Reviewer #1: This is an important topic of the life cycle because adolescence as stated by the authors is the stage where physical, psychosocial, and cognitive maturity are largely accomplished and the dramatic growth increases nutrient needs, making the study of food variety and consumption in nutritional status crucial.

There are several points that need to be clarified in a major revision to improve the paper, particularly:

1) Introduction – this area needs to be further developed (expand the literature review) with more articles or studies centered on adolescents’ food variety and consumption scores. You can include more information about the evidence that exists around the relevance/impact of food variety and consumption score in adolescent nutritional status, health, and growth. It would be good to see more description of the studies conducted around this with respect to adolescents specifically than in general (using adults, pregnant women, etc.) and then describe the gap with respect to food variety and consumption score in adolescents in Africa/Ghana.

You can expand the food consumption score also which is only 4 lines on a stand-alone paragraph.

2) Materials and methods –

• The authors will have to describe in detail the study area. In the topic, “Rural District” is used, which immediately causes the reader to assume the study was conducted in mainly a rural area. However, in the Abstract and Results, peri-urban residency has been used, which is not a rural area because it is a different classification on its own. The authors should be clear about what factors were taken into consideration in classifying the areas the study took place. This is connection with this statement in the Discussion – “Most of the rural settlers send their produce to urban and peri-urban areas for sale on market days, and this could have contributed to the increased food security situation among participants living in these areas.”

• The authors should explain why the exclusion was made in this sentence, especially with the adolescence stage associated with poor food choices due to increased autonomy and more time spent out of the home with peers. “The food variety score was calculated from 63 selected food items. The list of foods from alcoholic and non-alcoholic beverages, sweets and snacks were excluded from the calculation of the food variety scores.”

• “Each food group was multiplied by a weight factor.” How was the determination of this weight factor done (explain) – what is the basis of assigning the numbers or what determines the number that a food group needs to be assigned?

•

3) Results –

• “A large proportion of participants were food secure; primary education (74.3%), having a guardian with formal education (78.6%), low-income status (75.7%) and not skipping meals (51.4%).” – The use of semicolon after the food secure in this sentence means it is related to the variables following it. This is incorrect for low-income status in relation to food security. Based on the literature, food security is associated with moderate to high-income status. The authors should revise the statement.

• Table 4: If there is nothing interesting to talk about for the partial correlation of the food consumption score, for example the negative values or non-significance, that column should be omitted.

4) Discussion –

• Add a reference to this sentence, “It is believed that people living in rural communities, especially farming communities, have access to a wide variety of foods compared to those in urban areas. Similarly, individuals with busy schedules may depend on fast foods which may be monotonous if they don’t take pains to plan their meals.

• The sentence above contradicts this statement, “The association between residency and the food consumption score is confirmed by a study among pregnant women in Ethiopia [21]. Results of a recent study in Burkina Faso also supports the findings that rural or urban settlement influences food consumption [20]. One’s geographical location affects the type of food items that can be grown, be available or be accessible, and these can influence the varieties of food that may be available to them.” The authors should rephrase this statement. Again, the definition of rural, peri-urban, urban in relation to the title of the manuscript will be helpful.

• The authors should give interpretation of the results of food consumption score with male gender.

5) Conclusion – this section should be re-written. It is just a repetition of the same information from the Results. The authors should in addition, provide the public health implication of the study here, the strengths and weaknesses or study limitations, etc.

6. PLOS authors have the option to publish the peer review history of their article (what does this mean?). If published, this will include your full peer review and any attached files.

Reviewer #1: No

---

## [Author Response · Author response to Decision Letter 0]

26 Dec 2022

Manuscript ID: PONE-D-22-28451

Response to Reviewers

Dear Sir/Madam,

Thank you for giving us the opportunity to submit a revised draft of the manuscript titled, " PREDICTORS OF FOOD VARIETY AND FOOD CONSUMPTION SCORES OF ADOLESCENTS LIVING IN A RURAL DISTRICT IN GHANA" for publication in PLOS ONE. We appreciate the time and effort that you and the reviewers dedicated to providing feedback on our manuscript and are grateful for the insightful comments on and valuable improvements to our paper. We have incorporated suggestions made by the reviewers. Those changes are highlighted within the revised manuscript. Please see below, for a point-by-point response to the reviewers’ comments and concerns. 

Reviewer(s)' comments to the Authors:

Reviewer: 1

Comments to the Author

1. Introduction – this area needs to be further developed (expand the literature review) with more articles or studies centered on adolescents’ food variety and consumption scores. You can include more information about the evidence that exists around the relevance/impact of food variety and consumption score in adolescent nutritional status, health, and growth. It would be good to see more description of the studies conducted around this with respect to adolescents specifically than in general (using adults, pregnant women, etc.) and then describe the gap with respect to food variety and consumption score in adolescents in Africa/Ghana.

You can expand the food consumption score also which is only 4 lines on a stand-alone paragraph.

Response: Thank you.

Additional information has been added introduction.

The paragraph 5 in the introduction now reads:

Adolescents living in food secure household have a reduced prevalence of poor nutritional status, negative consequences of undernutrition on cognitive development and mental health disorders such as poor mood, anxiety, depression and behavioural problems [27, 28]. This psychological distress transcends into adulthood [29]. Food insecurity has also been documented to pose health challenges such as poor dietary pattern, obesity, asthma and increased the health care cost for adolescents [30-32].

2. Materials and methods –

• The authors will have to describe in detail the study area. In the topic, “Rural District” is used, which immediately causes the reader to assume the study was conducted in mainly a rural area. However, in the Abstract and Results, peri-urban residency has been used, which is not a rural area because it is a different classification on its own. The authors should be clear about what factors were taken into consideration in classifying the areas the study took place. This is connection with this statement in the Discussion – “Most of the rural settlers send their produce to urban and peri-urban areas for sale on market days, and this could have contributed to the increased food security situation among participants living in these areas.”

Response: Thank you

‘Rural district’ differs from ‘Rural Ghana’ or ‘rural communities.’

The rural district does not entirely mean a cluster of rural areas. What it means is that majority (55-65%) of the areas under this district is rural. Development is also very low in those areas. This particular district is very far from the regional capital, Kumasi. 

This district has a capital with better or more amenities or infrastructure compared to the other areas in the district hence it is classified as peri-urban by the Asante-Akim South Municipal Assembly. The district capital cannot also be compared to the regional capital in terms of development, education, health amenities etc. 

In Ghana, most of the farmers in the rural areas send their produce to the urban and peri-urban areas for sales. This will enable the farmers to buy other items that are not available in the area where they live.

Additional information has been added to the study design, area and participants (paragraph 1) under the materials and method as recommended. It now reads:

Study design, area and participants

This cross-sectional study was conducted at the Asante-Akim South municipality in the Ashanti Region of Ghana. It is one of the rural districts in Ghana with farming as the major occupation of the people. The district has land size of 0.5% of the total land in Ghana. The food crops largely cultivated are cassava, plantain, cocoyam, yam, rice and maize. About 83% of the population lives in the rural areas of the district while rest reside within the peri-urban zone of the district [33, 34]. 

• The authors should explain why the exclusion was made in this sentence, especially with the adolescence stage associated with poor food choices due to increased autonomy and more time spent out of the home with peers. “The food variety score was calculated from 63 selected food items. The list of foods from alcoholic and non-alcoholic beverages, sweets and snacks were excluded from the calculation of the food variety scores.”

Response: Thank you

Reasons for the exclusion: The food list for alcoholic and non-alcoholic beverages (such as ‘pito’, palm wine, coffee etc), sweets and snacks exempted because they were rarely or never consumed. Also, most of the participants did not provide response to the foods under this section.

Additional information has been added to the food variety score under the materials and methods. It now reads:

Food variety score

The food variety score was calculated from 63 selected food items. The list of foods from alcoholic and non-alcoholic beverages (such as ‘pito’, palm wine and coffee), sweets and snacks were excluded from the calculation of the food variety scores. The list of foods exempted were rarely or never consumed. Food item consumed at least once a week was scored ‘1’ and ‘0’ for those not consumed at all. The scores were summed to obtain the food variety score. The food variety score (FVS) was ranked ‘high’ (≥26) or ‘low’ (≤25) based on the mean food variety scores.

• “Each food group was multiplied by a weight factor.” How was the determination of this weight factor done (explain) – what is the basis of assigning the numbers or what determines the number that a food group needs to be assigned?

Response: Thank you.

The weight factors and its assignment to food groups were adopted from the World Food Program document (The full paper or document can be found in reference 17 in the manuscript). It is an acceptable method adopted and used by other researchers. 

3. Results –

• “A large proportion of participants were food secure; primary education (74.3%), having a guardian with formal education (78.6%), low-income status (75.7%) and not skipping meals (51.4%).” – The use of semicolon after the food secure in this sentence means it is related to the variables following it. This is incorrect for low-income status in relation to food security. Based on the literature, food security is associated with moderate to high-income status. The authors should revise the statement.

Response: Thank you

The semicolon has been deleted. 

In terms of low-income status and food security percentage, the reviewer is encouraged to kindly make time to read from table 1.

Additional information has been added, the sentence now reads:

A large proportion of participants in the food secure category had primary education (74.3%), guardian with formal education (78.6%), low-income status (75.7%) and did not skip meals (51.4%).

• Table 4: If there is nothing interesting to talk about for the partial correlation of the food consumption score, for example the negative values or non-significance, that column should be omitted.

Response: Thank you

Additional information has been added to the results description of table 4 (paragraph 2). It reads:

The partial correlation showed a weak positive relationship between food consumption score and most of the dietary vitamins and minerals. An inverse weak correlation was found between food consumption score and pyridoxine, vitamin C and sodium. None of the relationship was significant (p<0.05) (Table 4). 

4. Discussion –

• Add a reference to this sentence, “It is believed that people living in rural communities, especially farming communities, have access to a wide variety of foods compared to those in urban areas. Similarly, individuals with busy schedules may depend on fast foods which may be monotonous if they don’t take pains to plan their meals.

Response: Thank you

The sentence is deleted from the paragraph two of the manuscript.

• The sentence above contradicts this statement, “The association between residency and the food consumption score is confirmed by a study among pregnant women in Ethiopia [21]. Results of a recent study in Burkina Faso also supports the findings that rural or urban settlement influences food consumption [20]. One’s geographical location affects the type of food items that can be grown, be available or be accessible, and these can influence the varieties of food that may be available to them.” The authors should rephrase this statement. Again, the definition of rural, peri-urban, urban in relation to the title of the manuscript will be helpful.

Response: Thank you

The referred or said sentence has been deleted from the manuscript.

5. Conclusion – this section should be re-written. It is just a repetition of the same information from the Results. The authors should in addition, provide the public health implication of the study here, the strengths and weaknesses or study limitations, etc.

Response: Thank you

The conclusion has been revised. It now reads: 

Conclusion 

The study showed that majority of adolescents had acceptable food consumption scores and high food variety scores. The income status of guardian had significant impact on the food variety score of adolescents. Factors such as gender, residency, frequency of meals per day and food variety score were significantly associated with food consumption scores. The predictors of food security based on the study outcome were adolescent males, peri-urban residency, three or more frequency of meals per day and high food variety score. Governments and non-governmental organization are encouraged to implement nutrition education programmes and interventions aimed at providing adolescents with adequate nutrition knowledge, and also empower guardians economically to increase their purchasing power as it will influence their wards to have access to variety of foods for consumption and possibly increase their food consumption scores. A well-nourished adolescent is a potential healthy adult. Nutrition specific and nutrition sensitive programmes aimed at promoting adolescent health should involve guardians. 

As a limitation to this study, the recall bias of the dietary intake methodology could affect the food consumption and variety scores. The subjective weight factor of the food consumption score may not be applicable for all food consumption patterns in all countries. The food consumption score is not able to show the seasonal changes in food and also measure the quantity of food gap.

Journal Requirement

Response: Thank you

Additional information has been added to the Ethics Section under the material and methods: it now reads

Ethics

The study protocol was approved by the Ethics Review Board of the Kwame Nkrumah University of Science and Technology, (reference number: CHRPE/AP/585/22), Ghana. Permission was granted by the Asante-Akim South Municipal Health Directorate. The purpose of the study was explained to the adolescents and guardians. Adolescents who gave their assent backed by their guardian’s consent either by signing or thumb printing on the informed consent form qualified to participate in the study. 

6. Some grammatical errors sighted have been corrected. 

Thank you in anticipation for a favourable response. 

Yours faithfully,

Dr Michael A. Wiafe.

---

## [Decision Letter · Decision Letter 1]

3 Apr 2023

PONE-D-22-28451R1PREDICTORS OF FOOD VARIETY AND FOOD CONSUMPTION SCORES OF ADOLESCENTS LIVING IN A RURAL DISTRICT IN GHANAPLOS ONE

Dear Dr. Wiafe,

Thank you for submitting your manuscript to PLOS ONE. After careful consideration, we feel that it has merit but does not fully meet PLOS ONE’s publication criteria as it currently stands. Therefore, we invite you to submit a revised version of the manuscript that addresses the points raised during the review process.

Both the reviewers have made some serious queries. While preparing the revision, please go over the comments and suggestions of reviewer 2  and incorporate the suggestions.

We look forward to receiving your revised manuscript.

Kind regards,

Muhammad Khalid Bashir, PhD

Academic Editor

PLOS ONE

Reviewers' comments:

Reviewer's Responses to Questions

**Comments to the Author**

1. If the authors have adequately addressed your comments raised in a previous round of review and you feel that this manuscript is now acceptable for publication, you may indicate that here to bypass the “Comments to the Author” section, enter your conflict of interest statement in the “Confidential to Editor” section, and submit your "Accept" recommendation.

Reviewer #1: All comments have been addressed

Reviewer #2: (No Response)

2. Is the manuscript technically sound, and do the data support the conclusions?

Reviewer #1: (No Response)

Reviewer #2: No

3. Has the statistical analysis been performed appropriately and rigorously? 

Reviewer #1: (No Response)

Reviewer #2: No

4. Have the authors made all data underlying the findings in their manuscript fully available?

Reviewer #1: (No Response)

Reviewer #2: No

5. Is the manuscript presented in an intelligible fashion and written in standard English?

Reviewer #1: (No Response)

Reviewer #2: No

6. Review Comments to the Author

Reviewer #1: (No Response)

Reviewer #2: Thank you for providing an opportunity to review this paper. I read this manuscript with interest but found a number of issues with it. In my opinion, this work requires special attention from the authors before resubmission. Please find my suggestions and comments as follows, and I hope these will assist authors in improving the quality of the paper.

General comments

The current form of the manuscript has issues in all its sections, starting with the introduction and ending with the conclusion. Both the methodology and the section were poorly written. Discussion is merely repeating the results.

Specific comments

1. Mention the country name in the abstract (method) section of the manuscript.

2. Please don't use the same words as the study's keywords, which are already in the title of the paper.

3. Please describe the gap in the prior literature and the contribution that this study intends to make. Additionally, please include the contribution of this study in the literature.

4. I could not find the hypotheses of the study.

5. I could not find the details of the study beneficiaries in the manuscript.

6. Why did you select the Asante-Akim South municipality in the Ashanti Region of Ghana? Please talk about how important it is for Ghana's food security. Also discuss the other characteristics of the study area.

7. How did you estimate the total sample size (137)? Did you use any sampling formula to reach this sample size?

8. Explain the sampling section criteria in more detail.

9. I am wondering about the time period during which food related information was obtained.

10. Please explain your questionnaire in your revised manuscript.

11. How did you check the reliability of the data collection instrument?

12. Explain in full detail how food scores were calculated.

13. Please explain how you decided to use binary logistic regression. Put its formulas together and explain the explanatory and dependent variables.

14. The ethics section could be taken out of the manuscript and placed with other statements, according to the journal's rules.

15. A subheading "descriptive statistics" should be added immediately after "Results."

16. The interpretation of descriptive statistics needs extensive revision. Please revise it. Look for high-quality papers to help you with your writing.

17. The legend in Table 1 should be revised as it does not present an association.

18. Please present the results of binary logistic regression in a proper way. Get help from relaxant literature. What is the reason for selecting only four variables in the logistic model? The interpretation of the model results is not enough.

19. How did you estimate micronutrients? No information is provided about it in the methodology section, or I am missing something (Table 4).

20. I could see a number of limitations with the study. Why did the author not include it in the manuscript?

21. Most of the discussion is repeating the results. Please discuss the results by repeating them.

22. The table data presentation is totally absurd. Improve the table data presentation.

7. PLOS authors have the option to publish the peer review history of their article (what does this mean?). If published, this will include your full peer review and any attached files.

Reviewer #1: No

Reviewer #2: **Yes: **Dr. POMI SHAHBAZ

---

## [Author Response · Author response to Decision Letter 1]

7 Apr 2023

Manuscript ID: PONE-D-22-28451R1

Response to Reviewers

Dear Sir/Madam,

Thank you for giving us the opportunity to submit a revised draft of the manuscript titled, " PREDICTORS OF FOOD VARIETY AND FOOD CONSUMPTION SCORES OF ADOLESCENTS LIVING IN A RURAL DISTRICT IN GHANA" for publication in PLOS ONE. We appreciate the time and effort that you and the reviewers dedicated to providing feedback on our manuscript and are grateful for the insightful comments on and valuable improvements to our paper. We have incorporated some of the suggestions made by the reviewers and also provided some rebuttals on the comments. Those changes are highlighted within the revised manuscript. Please see below, for a point-by-point response to the reviewers’ comments and concerns. 

Reviewer(s)' comments to the Authors:

Reviewer: 1

Comments to the Author

1.Mention the country name in the abstract (method) section of the manuscript.

Response: Thank you.

Additional information has been added introduction.

The country’s name has been mentioned in the abstract as suggested. It is highlighted in the manuscript

2.Please don't use the same words as the study's keywords, which are already in the title of the paper.

Response: Thank you

keywords are words provided to make the article easily discoverable and must also be related to the work. Among the keywords already provided, the authors want to know which of them are outside the domain of the article or the work?

The reviewer(s) can suggest alternative keywords and if they worth being used as keywords by the authors we will fix it/them in the manuscript.

3.Please describe the gap in the prior literature and the contribution that this study intends to make. Additionally, please include the contribution of this study in the literature.

Response: Thank you

The last paragraph of the introduction addresses the comments 3. There is paucity of knowledge about food variety and food consumption scores of adolescents in rural communities in Ghana. Additional information has also been added to the last paragraph of the introduction.

4.I could not find the hypotheses of the study.

Response: Thank you.

5.I could not find the details of the study beneficiaries in the manuscript.

Response: Thank you.

Information about the study beneficiaries are provided under the study design, area and participants of the materials and method section. 

6. Why did you select the Asante-Akim South municipality in the Ashanti Region of Ghana? Please talk about how important it is for Ghana's food security. Also discuss the other characteristics of the study area.

Response: Thank you

The reasons for the selection of the Municipality is based on the title (living in rural district) and dwellings of the percentage (83%) of the population.

The information under the study design, area and participants of the materials and methods section clearly explains the comment 6.

The reviewer(s) should kindly indicate the specific characteristics of the study area that need to be added .

7. How did you estimate the total sample size (137)? Did you use any sampling formula to reach this sample size?

Response: Thank you

The sample size used for this study was that calculated for an intervention study. The data used for this paper was collected as part of baseline data of an intervention study. For the calculation of the sample size, assess the published protocol via https://doi.org/10.1177/02601060221074433

8. Explain the sampling section criteria in more detail.

Response: Thank you

Additional information has been added to the sampling technique in the manuscript. It now reads: 

A total of one hundred and thirty-seven (137) healthy adolescents aged 10-14 years were selected using a multi-stage sampling method from the municipality for this study.

9. I am wondering about the time period during which food related information was obtained.

Response: Thank you

Since this study is cross-sectional all information were collected once and at the same time.

10. Please explain your questionnaire in your revised manuscript.

Response: Thank you

The authors would want to know how the reviewer(s) want the questionnaire to be explained?

For instance: 

1)how many times do you eat a day? 1. once 2. twice

2)Age: …………..

3)Sex: 1. male 2. female

4)Do you skip any meal? 1. Yes 2. No

11. How did you check the reliability of the data collection instrument?

Response: Thank you

A pilot study was conducted with the questionnaire before the actual data collection.

12. Explain in full detail how food scores were calculated.

Response: Thank you

Additional information has been added to the calculations of the food scores in the manuscript.

13. Please explain how you decided to use binary logistic regression. Put its formulas together and explain the explanatory and dependent variables.

Response: Thank you

Binary logistic regression is a regression model where the target variable is binary, that is, it can take only two values, (0 or 1) or (Yes or No). The analysis was not done manually hence provision of the formula may not be necessary. SPSS software was used in the data analysis.

The independent variables or predictors were sex, community, frequency of meals per day and food variety status. The dependent or outcome variable was food secure. 

14. The ethics section could be taken out of the manuscript and placed with other statements, according to the journal's rules.

Response: Thank you for bringing it to our attention

15. A subheading "descriptive statistics" should be added immediately after "Results."

Response: Thank you

Recommendation has been implemented in the manuscript. 

Other sub-headings have also been introduced in the results to maintain consistency. 

16. The interpretation of descriptive statistics needs extensive revision. Please revise it. Look for high-quality papers to help you with your writing.

Response: Thank you

Some revisions have been made as requested in the descriptive statistics. The authors request the reviewer(s) to be kind to indicate any other specific revision. 

17. The legend in Table 1 should be revised as it does not present an association.

Response: Thank you

The authors do not agree entirely. The chi-square test is a test of association. 

A revision has been made in the heading for Table 1. 

18. Please present the results of binary logistic regression in a proper way. Get help from relaxant literature. What is the reason for selecting only four variables in the logistic model? The interpretation of the model results is not enough.

Response: Thank you

The position of the constant (1) has been changed in Table 3. 

The authors wanted to know whether the same factors that produced significant relationship between food consumption score were also able to significantly predict food security. Hence the four variables.

19. How did you estimate micronutrients? No information is provided about it in the methodology section, or I am missing something (Table 4).

Response: Thank you bring this to our attention

Additional information has been added to the dietary intake under the materials and methods sections. It reads:

A 24-hour recall was used to collect three days (one weekend and two working days) dietary data. Participants were asked to tell us how many times (breakfast, mid-morning snack, lunch, mid-afternoon snack, dinner and bed-time snack) they eat a day and the detailed of the food consumed. Participants were shown food handy measures to estimate how much of the food consumed at each meal. Each food was converted to grams and the nutrients were estimated using the Nutrient Analysis Template developed by the University of Ghana. The averages for the nutrients were recorded.

20. I could see a number of limitations with the study. Why did the author not include it in the manuscript?

Response: Thank you

Currently, the few limitations identified were the ones written in the manuscript. The authors are open to further suggestions on limitations by the reviewer(s). 

21. Most of the discussion is repeating the results. Please discuss the results by repeating them.

Response: Thank you

In discussion, it is expected that the main outcome(s) of the current study is stated and compared or contrast with other studies findings. Probably given reasons for the otherwise of contrary outcomes between the current and the previous or other(s) studies. 

The authors did exactly the suggestion above in the manuscript, however the authors are ready to accept suggestion(s) from the reviewer(s) that will add value to the discussion.

Additional information has been added to the discussion and highlighted. 

22. The table data presentation is totally absurd. Improve the table data presentation. 

Response: Thank you 

The reviewer(s) must be specific about the specific table(s) that is/are not well presented, and the authors will be ready to address it.

Some grammatical errors sighted have been corrected. 

Thank you in anticipation for a favourable response. 

Yours faithfully,

Michael A. Wiafe, PhD.

---

## [Decision Letter · Decision Letter 2]

17 May 2023

Predictors of Food Variety and Food Consumption Scores of Adolescents Living in a Rural District in Ghana

PONE-D-22-28451R2

Dear Dr. Wiafe,

We’re pleased to inform you that your manuscript has been judged scientifically suitable for publication and will be formally accepted for publication once it meets all outstanding technical requirements.

Kind regards,

Muhammad Khalid Bashir, PhD

Academic Editor

PLOS ONE

Additional Editor Comments (optional):

Reviewers' comments:

Reviewer's Responses to Questions

**Comments to the Author**

1. If the authors have adequately addressed your comments raised in a previous round of review and you feel that this manuscript is now acceptable for publication, you may indicate that here to bypass the “Comments to the Author” section, enter your conflict of interest statement in the “Confidential to Editor” section, and submit your "Accept" recommendation.

Reviewer #2: All comments have been addressed

2. Is the manuscript technically sound, and do the data support the conclusions?

Reviewer #2: Partly

3. Has the statistical analysis been performed appropriately and rigorously? 

Reviewer #2: Yes

4. Have the authors made all data underlying the findings in their manuscript fully available?

Reviewer #2: No

5. Is the manuscript presented in an intelligible fashion and written in standard English?

Reviewer #2: Yes

6. Review Comments to the Author

Reviewer #2: (No Response)

7. PLOS authors have the option to publish the peer review history of their article (what does this mean?). If published, this will include your full peer review and any attached files.

Reviewer #2: **Yes: **Dr. Pomi Shahbaz

---

## [Editor Report · Acceptance letter]

22 May 2023

PONE-D-22-28451R2 

Predictors of Food Variety and Food Consumption Scores of Adolescents Living in a Rural District in Ghana 

Dear Dr. Wiafe:

I'm pleased to inform you that your manuscript has been deemed suitable for publication in PLOS ONE. Congratulations! Your manuscript is now with our production department. 

Kind regards, 

on behalf of

Dr. Muhammad Khalid Bashir 

Academic Editor

PLOS ONE